# Factors Influencing COVID-19 Vaccination among Primary Healthcare Nurses in the Pandemic and Post-Pandemic Period: Cross-Sectional Study

**DOI:** 10.3390/vaccines12060602

**Published:** 2024-05-31

**Authors:** Zorica Pristov, Bojana Lobe, Maja Sočan

**Affiliations:** 1Community Health Centre Ljubljana, 1000 Ljubljana, Slovenia; zorica.pristov@zd-lj.si; 2Faculty of Social Sciences, University of Ljubljana, 1000 Ljubljana, Slovenia; bojana.lobe@fdv.uni-lj.si; 3National Institute of Public Health, 1000 Ljubljana, Slovenia

**Keywords:** primary care nurse, COVID-19, vaccine, booster dose, survey

## Abstract

The decision to vaccinate against COVID-19 is primarily a personal choice influenced by numerous factors. Vaccine acceptance and a positive attitude towards vaccination among nurses have an impact on patients’ willingness to vaccinate. To assess COVID-19 vaccination coverage among primary healthcare nurses and to associate socio-demographic factors, comorbidity, self-rated health, and unhealthy lifestyle with the decision to be vaccinated, we conducted an online cross-sectional study from March to May 2023 using a self-administrated questionnaire. Probability sampling was used to select 32 health centers and nurses were invited via email. Among the 560 participants who completed survey, 78.3% and 50.8% received the primary two-dose course and at least one booster dose of COVID-19 vaccine, respectively. Primary care nurses who were ≥41 years of age, physically less active, and those who were overweight opted statistically significantly more often for the primary vaccination scheme (*p* = 0.00, 0.015 and 0.017, respectively). Education and the living environments of primary care nurses did not significantly influence the decision to receive two primary COVID-19 doses. Likewise, good self-rated health and comorbidity did not contribute significantly to the vaccination decision. Nurses that were vaccinated with booster doses were significantly more often overweight (*p* = 0.034) and ≥41 year of age (*p* = 0.000).

## 1. Introduction

The COVID-19 pandemic fundamentally intruded upon society and altered people’s lives. During the pandemic waves induced by various SARS-CoV-2 variants, there was an increase in morbidity, a burden on the healthcare system, and a rise in mortality, particularly pronounced among older adults. While excess mortality was especially evident among older adults (≥85), there were also more deaths among those aged ≥65 and individuals in the middle-aged bracket. By the end of 2020, the International Council of Nursing reported at least 1500 deaths among nurses and highlighted an underreported death toll, with reports coming from only 44 countries [1]. The World Health Organization (WHO) estimated that around 100,000 healthcare workers died in the first 15 months of the pandemic [2]. Despite the successful containment of the pandemic, the virus continues to circulate, and the WHO regularly updates its vaccination recommendations against COVID-19, still advocating vaccination among the general population and healthcare workers [3].

During the first year of the pandemic, new technologies facilitated the rapid development of vaccines against COVID-19. There were opinions that the vaccines were developed swiftly, perhaps too hastily, to adequately study their adverse effects and ensure vaccine safety [4]. The production and distribution of COVID-19 vaccine doses were inadequate, reducing vaccine accessibility and hindering vaccination efforts. Healthcare workers were among the first offered COVID-19 vaccination, yet readiness to vaccinate was mixed and, in many countries, below expectations. Severe but rare adverse effects following vaccination with vector vaccines intensified hesitancy in both the general and professional public, undermining the position that COVID-19 vaccines were unconditionally safe [5,6,7].

The factors leading to vaccination hesitancy are varied, and can stem from the individual’s personality, socioeconomic determinants, risk perception, disease severity, and social environment. Contextual influences encompass socio-cultural, environmental, healthcare system, economic, and political factors, including trust in the health authorities’ recommendations and the healthcare system. Vaccine type and vaccination, vaccine characteristics, development, and the vaccination process fall into a distinct group of determinants [8].

Attitudes towards vaccination, including vaccination against COVID-19, among nurses in primary healthcare is a poorly researched area. The inhibiting and enabling factors for vaccination within this professional group, who are in direct contact with patients, carry out vaccination programs for children and adults, and significantly influence people’s decisions, are not well studied.

In meta-analyses of papers addressing the vaccination of healthcare workers against COVID-19, studies focusing on the vaccination of nurses in primary healthcare were omitted; instead, the entire professional group was researched. After reviewing 51 articles, Khubchandani et al. concluded that nurses were less likely to decide to vaccinate against COVID-19 than physicians [5]. Furthermore, nurses shared the same concerns about vaccine safety and side effects as the general population. They also found that even fewer healthcare workers opted for the booster dose than the primary doses. In their review of 71 articles on the acceptance of COVID-19 vaccination among healthcare workers, Wang et al. found that physicians were more inclined to vaccinate against COVID-19 than nurses, believing that those who chose not to vaccinate significantly influenced patients’ similar decisions [9]. A meta-analysis conducted by Galanis et al. that analyzed articles involving nurses confirmed that this professional group is less likely to choose vaccination than physicians [7].

Our study aimed to investigate the determinants of vaccination coverage within the poorly studied subgroup of nurses working in primary healthcare clinics. The study was conducted towards the end of the pandemic, as the WHO declared on 4 May 2023 that COVID-19 was no longer a public health emergency of international concern (PHEIC) [10]. This fact adds value to the study of this professional group’s attitude towards future COVID-19 vaccinations and non-mandatory vaccinations against other diseases during possible future pandemics.

The World Health Organization (WHO) defines socio-demographic characteristics as factors influencing health. The decision to vaccinate can be linked to some socio-demographic factors such as age, gender, education, living environment, etc. [11]. According to the Health Belief Model, health risk is a decision-making factor. Janz and Becker calculated significance ratios indicating how often each factor was statistically significant in 46 studies. The significance values of the impact of susceptibility to infection, seriousness of the disease, perceived benefits, and severity on vaccine acceptance were 81%, 65%, 78% and 89%, respectively [12].

In our study, we aimed to examine the critical enabling and impeding factors in the decision to vaccinate and to determine the explanatory power of variables such as associated diseases, factors of an unhealthy lifestyle, and self-rated health among nurses at the primary level. Our research goal was to identify the key demographic, health, and lifestyle factors influencing the decision to receive primary and booster vaccinations against COVID-19 among nurses in health centers.

## 2. Materials and Methods

### 2.1. Research Design and Sampling

We incorporated a cross-sectional survey design, using a sample of nurses employed at health centers across Slovenia. There are approximately 6000 individuals employed in nursing in these centers. We selected at least half of the health centers from each statistical region in the country. Consequently, in the initial stage, using probability sampling via the simple random sampling method (drawing lots), 32 health centers were chosen out of 66, representing 3130 nurses. Nursing managers in health centers were requested to forward an invitation via email and invite nurses to participate in the survey, linking them to the online questionnaire. All participants were nurses, with no discrimination for educational level or the type of work they performed within the health center framework. The survey did not include nursing students on practical training in health centers or those temporarily employed.

### 2.2. Data Collection Procedure

The survey was conducted from 20 March to 4 May 2023, involving randomly selected health centers. A follow-up reminder was sent 14 days after the initial invitation. Before the study, a pilot survey was conducted in January 2023 to confirm the questionnaire’s adequacy.

### 2.3. Measurement Instrument

We utilized a standardized and structured questionnaire predominantly composed of closed-type questions. The questionnaire contained socio-demographic queries and assertions to gauge attitudes (incorporating five-level Likert statements, which sometimes formed a Likert scale), and was integrated with the Health Belief Model and the Theory of Planned Behavior [13]. Additional questions covered self-rated health, associated diseases, lifestyle self-assessment, and decision-making regarding vaccination.

### 2.4. Variables in Question Blocks Considered in Four Models

The data collected on the decision to vaccinate or not to receive vaccination against COVID-19 were associated with the following:-Age;-Level of education;-Living environment;-Associated diseases (asthma or other chronic respiratory disease, chronic heart disease, diabetes, chronic kidney or liver disease, other chronic health issues, or immunosuppression due to illness/medication); -Unhealthy lifestyle factors (smoking, excessive body weight, physical activity less than 150 min per week, and consumption of less than 400 g of fruits and vegetables daily).

We formed four research questions to ascertain the connections between and the influence strength on the decisions to vaccinate against COVID-19 with the primary and booster schemes:Among which demographic variables (age, education level, and living environment) were the decision to vaccinate against COVID-19 higher?Are individuals with one or more associated diseases more likely to decide to vaccinate against COVID-19?Do individuals who consider themselves healthy more frequently opt to vaccinate against COVID-19?Are individuals with unhealthy lifestyle factors (smoking, excessive body weight, physical activity less than 150 min per week, and less than 400 g of fruits and vegetables daily in the diet) more inclined to vaccinate against COVID-19?

### 2.5. Data on Study Participants

We invited 3130 nurses to participate in the web-based anonymous survey. We noticed that 913 nurses opened the survey but not all of them took the decision to participate. Some nurses completed the questionnaire inadequately and were omitted from the analysis. There were 560 questionnaires adequately completed and available for the final analysis.

### 2.6. Data Analysis Procedures and Methods

Descriptive statistics, the Chi-square test for the association between two variables, the Chi-square test of equal probability, and Cramer’s V test were utilized for the verification process.

We used logistic regression to analyze the data. The dependent variables were binary (whether the individual was vaccinated with the basic scheme or not and whether the individual was vaccinated with the booster dose or not). We estimated the probability to get vaccinated based on independent variables (socio-demographic data, health conditions, and lifestyle). Maximum likelihood was used to estimate the coefficients of the regression models. The statistical significance of the coefficients was checked using the Wald test, using a value of *p* < 0.05 as the significance threshold. The coefficients were interpreted using an exponent (Exp(B)) representing the likelihood ratio.

Data analysis was performed using the statistical software package SPSS (version 29.0).

## 3. Results

### 3.1. Descriptive Analysis of Demographic Data

Of the 560 respondents, 505 were women (90.5%) and 53 were men (9.5%). Two did not answer the gender question. The average age of the respondents was 43.6 years (ranging from 21 to 64 years). Thirteen participants (2.3% of 560) did not respond regarding age. Fourteen out of five hundred and sixty individuals (2.5%) did not answer the question regarding their achieved level of education. Of the 546 respondents who did answer the education level question, approximately one-third (180 persons, 33%) had completed secondary school, while the remainder had completed higher vocational school or postgraduate (master’s/doctoral) studies (366 persons, 67%).

Just under half of the respondents indicated living in rural areas (237 persons, 42.3%), 176 (31.4%) resided in urban settings, and a quarter were from suburban areas (147 persons, 26.3%).

### 3.2. Analysis of Decision-Making Regarding COVID-19 Vaccination

During the primary vaccination scheme against COVID-19 (two doses of mRNA vaccines Pfizer-Comirnaty^®^ or Moderna-Spikevax^®^, or two doses of the vector vaccine Vaxzevria^®^), 439 (78.3%) of the nurses were vaccinated, while a quarter never decided to get vaccinated. Fewer nurses (285 persons, 50.8%) opted for the booster dose. In Slovenia, only mRNA vaccines Pfizer-Comirnaty^®^ or Moderna-Spikevax^®^ were used for booster vaccination. The questionnaire did not collect data on how many nurses were primarily vaccinated with an mRNA or vector vaccine (Table 1 and Table 2).

There were no statistically significant differences in the decision to vaccinate against COVID-19 with either the primary or booster dose when considering gender and level of education. However, a greater proportion of primary healthcare nurses aged 41 years or older chose to be vaccinated with both the primary and booster doses. The percentages of vaccinated individuals residing in urban, suburban, and rural environments was comparable to the non-vaccinated nurses. Similarly, the portion of those who opted for the booster dose did not show any statistically significant differences concerning urban, peri-urban, or rural environments.

Of the nurses, 123 (21.9%) reported having at least one chronic health issue, with very few having at least two comorbidities (17 nurses, 3.0%). There were no differences in vaccination coverage among the nurses based on the absence or presence of a chronic disease or the self-assessment of health as good or poor (applicable to both primary and booster vaccinations). The proportion of vaccinated and unvaccinated individuals was also similar among those with asthma, diabetes, chronic liver or kidney disease, those immunocompromised due to medication or illness, and those with other chronic health issues. The only statistically significant difference was among nurses with chronic heart disease concerning the booster dose—those with heart conditions were more likely to choose the booster vaccination. In contrast, no difference was noted for the primary vaccination. The number of subjects with specific comorbidities was small, which may have led to a lack of statistical significance.

The proportions of individuals vaccinated with primary and booster doses who exhibited unhealthy lifestyle habits (insufficient fruit and vegetable intake, inadequate physical activity), obesity, or smoking were statistically significantly different among those who reported having excessive body weight (BMI over 30) and those who considered themselves to be insufficiently physically active, albeit only for the primary vaccination.

The strength of the association between certain variables was measured using Cramer’s V coefficient. We were interested in the association between the decision to vaccinate with the primary doses and selected variables (Table 3) and the association between the decision to vaccinate with booster doses and selected variables (Table 4). The associations between primary vaccination and factors such as age, excessive body weight, and insufficient physical activity were weakly statistically significant. The relationship between age and the booster dose of the COVID-19 vaccine was statistically significant and moderately associated. However, the association between excessive body weight and vaccination was statistically significant but weak.

Vaccination coverage with the booster dose decreased compared to the primary vaccination series. A more significant decline in the decision to receive booster doses was observed among women; however, it should be noted that the proportion of men was very small.

In the selected sample, a small number of nurses had comorbidities. Despite the small numbers, it was observed that the ratio of vaccinated to unvaccinated individuals with individual comorbidities shifted in favor of non-vaccination when deciding on the booster dose. In other words, nurses with asthma, other chronic respiratory diseases, renal issues, liver issues, diabetes, immune disorders, or other chronic health issues were less likely to choose vaccination, even though the risk of a more severe course of COVID-19 was higher in these individuals.

### 3.3. Multiple Regression Model for Basic Vaccination Scheme and Booster Dose

For the basic vaccination scheme, age was the most important variable, as with each additional year of age, the probability of vaccination increased by a factor of 2.446 (B = 0.894; Sig. < 0.001) (Table 5). Physical activity also had a positive influence, with individuals who were less physically active having a 1.911 times higher probability of vaccination (B = 0.648; Sig. = 0.045). Other variables, such as gender, education, environment of permanent residence, and the presence of other chronic diseases, did not have a statistically significant effect on the decision to be vaccinated.

For the booster dose, age was also the most important factor, as the probability of vaccination increased by a factor of 2.810 with each additional year of age (B = 1.033; Sig. < 0.001) (Table 6). Gender was also a significant factor, as men were 2.101 times more likely to receive a booster dose (B = 0.742; Sig. = 0.028). The presence of chronic heart disease increased the probability of vaccination by a factor of 5.096 (B = 1.628; Sig. = 0.030). Dietary habits, especially the consumption of less than 400 g of fruits and vegetables per day, had a negative effect on the decision to vaccinate, decreasing the probability of receiving the booster dose by a factor of 0.583 (B = −0.539; Sig. = 0.046). Other variables in the model did not have a statistically significant effect on the decision to receive the booster dose.

## 4. Discussion

In the initial phase of COVID-19 vaccination, when the pandemic and the spread of SARS-CoV-2 were at their highest and the burden on healthcare due to treatment and screening testing was considerable, Souza et al. conducted a qualitative study to determine nurses’ attitudes towards the vaccination campaign [14]. They were found to have a positive attitude towards vaccination, but faced challenges with the anti-vaccination movement.

Attitudes towards accepting COVID-19 vaccination have been the subject of many studies since the beginning of the pandemic, even before the development of vaccines. Research in 2020 and early 2021 investigated the attitude and readiness for COVID-19 vaccination among the general population and specific groups, such as healthcare workers [15,16,17]. Early studies found an association between higher education, the male gender, and older age with a greater inclination towards COVID-19 vaccination. In addition to research focused on the general population, the readiness of healthcare workers to vaccinate with COVID-19 vaccines was also examined. Biswas et al. reviewed 35 studies from various countries and found that, on average, 22.5% of 76,417 healthcare workers worldwide were undecided about COVID-19 vaccination [18]. Their main concerns (safety, efficacy, and side effects) were very similar to those of the wider public.

Studies examining the willingness to vaccinate among healthcare workers confirmed a greater propensity for vaccination among doctors than nurses [19,20]. It appeared that older healthcare workers who were usually more educated, especially doctors, and those who felt more threatened and trusted the state and institutional recommendations were more inclined towards vaccination.

Some studies before vaccine availability and in the initial period of COVID-19 vaccination focused on nurses and healthcare assistants and examined vaccination intent [9,21,22,23]. In 2022 and 2023, several studies emerged about actual vaccination coverage and factors influencing the decision to vaccinate among nurses and healthcare assistants, including views on mandatory COVID-19 vaccination [24,25]. From research published by Khubchandani et al., we obtained some significant findings about the vaccination decision-making of nurses and healthcare assistants that can be compared to ours [5]. However, their research included nurses and healthcare assistants at all levels of healthcare, not just primary. Factors that conditioned an acceptance of vaccination were male gender, chronic diseases, contact with COVID-19 patients, older age, greater perceived susceptibility to COVID-19 infection, and the desire to protect others [23,26]. Rabi et al. added trust in vaccine efficacy among respondents, and Patelarou et al. added trust in experts and the government [27,28]. The decision for booster vaccination among nurses and healthcare assistants has already been investigated by Viskupovič and Wiltze [29], as well as Galanis et al. [30].

After primary COVID-19 vaccination, the effectiveness of protection against COVID-19 declines. For at-risk groups and healthcare workers, booster doses are recommended to prevent infection and symptomatic disease. The studies showed reduced readiness for vaccination against COVID-19 with the first booster dose in healthcare workers [31,32]. A higher level of hesitancy was observed among nurses compared to physicians. Positive attitudes towards COVID-19 booster vaccination among nurses has an important impact on the acceptance of vaccination by the general population. The number of studies aiming to evaluate nurses’ attitudes towards COVID-19 vaccine boosters is limited. Galanis and et al. found that one-third of nurses were hesitant towards a second booster dose or new COVID-19 vaccines. The predictors of hesitancy were good self-perceived health, lack of comorbidities, lower level of education, and no flu vaccination in the previous season [30]. The efficacy, benefit, and necessity were the most frequent concerns regarding the booster doses. Perceived risk and a perceived severity of health risk with being vaccinated with a booster dose lower the willingness to get vaccinated [33]. A Greek study found a correlation between burnout due to the COVID-19 pandemic in nurses and negative attitudes towards booster doses of the COVID-19 vaccine. Strong social support mitigated the negative effect of occupational stress and buffered the negative relation between COVID-19 burnout and booster vaccination [34]. Similar to previous studies, this study confirmed that male gender, increasing age, and having a comorbidity were independent demographic predictors of booster vaccine intention among nurses [34].

Our study found that more men than women, older individuals aged 41 years or above, those with more than secondary education, and those living in suburban areas chose the booster dose compared to the primary vaccination. Among those who indicated having any comorbidity, those with chronic heart disease were most likely to opt for the booster vaccination, and least likely were those with asthma or other chronic respiratory diseases; having two or more comorbidities did not condition a greater decision to vaccinate, while those with one comorbidity were more inclined towards the booster dose. In terms of unhealthy lifestyle factors, excessive body weight (BMI over 30) was a condition leading to a higher decision for the booster dose, while a diet with insufficient fruits and vegetables (less than 400 g per day) was the least influential. Even those who self-identified as healthy were less likely to choose booster vaccination. The most significant declines in choosing booster vaccination against COVID-19 (30 to 38%) were among those who declared having asthma or other chronic respiratory diseases, those with a diet insufficient in fruits and vegetables (less than 400 g per day), those with two or more comorbidities, smokers, diabetics, those under 40 years of age, those living in urban areas, and physically inactive individuals.

Conversely, the smallest declines in choosing booster vaccination against COVID-19 (18.8 to 22.8%) were among those who stated they were 41 years or above, had one comorbidity, a chronic health issue not listed, chronic heart disease, or were male. Comorbidity is a significant risk for a more severe course of COVID-19. According to studies, 1.7 billion people worldwide (22% of the population) have at least one comorbidity associated with an increased risk of developing severe COVID-19. Our results are concerning as the nurses with comorbidities from our study forwent booster vaccinations.

Statistically significant differences concerning the decision to vaccinate following the primary scheme were observed in relation to age and unhealthy lifestyle, manifested as excessive body weight (BMI over 30) and insufficient weekly physical activity. In deciding on booster doses, statistically significant differences were found in relation to age and excessive body weight (BMI over 30).

Overall, there was a 28.3% decline in vaccination with the booster dose compared to the primary vaccination, meaning nearly one-third of those who received primary COVID-19 vaccination did not opt for the booster dose. The decline in vaccination uptake could be attributed to the improved epidemiological situation, reduced virulence of the virus, and new knowledge about this infectious disease.

This cross-sectional survey was conducted when the COVID-19 pandemic was subsiding and no longer posed the same threat as at its onset. With changing virus strains, attitudes towards vaccination also shifted. However, this research in Slovenia has, for the first time, gathered data on factors influencing the vaccination decision among nurses in primary care.

There are some limitations of the study. Firstly, the response rate was less than expected. The primary care nurses were invited twice to participate in the study but still the response rate was less than 20%. The low response rate might suggest a questionable representativeness and is one of the limitations of the study. Self-reporting is a common approach for collecting data in public health research. Self-reported data can be unreliable and are threatened by self-reporting bias. Bias can arise from social desirability and issues with recall periods, among other reasons [35]. The participants in the present study could have delivered erroneous responses depending on their ability to recall past events, e.g., vaccination with the COVID-19 vaccine. As the study population consisted of nurses, we believe that the probability of mistakenly remembering or incorrectly reporting vaccination status was highly unlikely, but still possible, and may have resulted in the incorrect classification of few cases in the groups analyzed. For most of the questions, the length of the recall period was short, which again lowers the probability of recall bias. Nevertheless, we are aware that overcoming recall bias can be difficult in practice. Self-reporting data can also be affected by an external bias caused by social desirability. We did not measure the internal or external validity of the data, which is one of the limitations of our study. The questionnaire was anonymous and the identification of participants was not possible, so we expect that their answers were trustworthy. 

In Slovenia, approximately 6000 individuals are employed in nursing within health centers in primary care. The number of adequately completed questionnaires was significantly lower than expected. This could be attributed to the waning pandemic, diminished interest in COVID-19, and a general decline in people’s willingness to participate in online surveys. One limitation of the study is the gender imbalance, with significantly more women employed in healthcare. Hence, a small proportion of men was anticipated. The gender imbalance in our study is similar to that in comparable studies, as there are significantly more women than men employed in healthcare, especially in the profession of nursing. The small number of men calls into question whether we can conclude that more male than female nurses opted for booster doses. This is an inherent obstacle as the gender imbalance cannot be solved, but our result is in line with similar studies that found that men were more willing to be vaccinated, regardless of education level.

Further in-depth quality studies are needed to look deeper into the reasons behind vaccine hesitancy among nurses through face-to-face interviews or focus groups to understand their concerns, beliefs, and attitudes towards COVID-19 vaccines and tackle cultural and contextual factors. There is a need to develop tailored educational programs for this professional group to address common misconceptions, provide accurate information about vaccine safety and efficacy, and highlight the importance of vaccination in protecting both individual and public health. Educational programs should be regularly evaluated for effectiveness, with vaccination rates analyzed before and after implementation. More research is warranted on the potential role of policy interventions in promoting workplace vaccination policies, offering incentives for vaccination, and addressing systemic barriers to vaccine access. An interesting concept at present is to encourage influential nurses to serve as vaccine advocates and role models within their communities to combat misinformation and build trust in vaccines through social networks. Lastly, developing effective strategies for addressing COVID-19 vaccine hesitancy among nurses and promoting vaccination as a critical public health measure should be recognized and supported by policymakers.

## 5. Conclusions

This study revealed that different factors are important in the decision to vaccinate against COVID-19 with the first dose and booster dose. Based on these findings, we can conclude that older primary care nurses are more in favor of being vaccinated against COVID-19. This applies to both the primary scheme and booster dose. It was also found that being less physically active increased the probability of vaccination with the basic scheme, while eating habits had a significant influence on booster dose uptake.

## Figures and Tables

**Table 1 vaccines-12-00602-t001:** Decision-making about vaccination against COVID-19 with primary and booster doses among nurses in primary healthcare in Slovenia according to gender, age, level of education, and living environment.

	Vaccinated with Primary Doses (*n* = 560)		Pearson Chi-Square Sig. (2-Sided)	Vaccinated with Booster Doses (*n* = 560)		Pearson Chi-Square Sig. (2-Sided)
	Yes	No		Yes	No	
	No. (%)	No. (%)		No. (%)	No. (%)	
Total	439 (78.4%)	121 (21.6%)		285 (50.9%)	274 (48.9%)	
Gender			0.597			0.375
Male	40 (75.5%)	13 (24.5%)		30 (56.6%)	23 (43.4%)	
Female	397 (78.6%)	108 (21.4%)		253 (50.2%)	251 (49.8%)	
Age			0.000			0.000
≤40 years	178 (70.1%)	76 (29.9%)		95 (37.4%)	159 (62.6%)	
≥41 years	251 (86%)	41 (14%)		184 (63.2%)	107 (36.8%)	
Education level			0.751			0.333
Secondary education	140 (77.8%)	40 (22.2%)		86 (48.0%)	93 (52.0%)	
Post sec. education	289 (79%)	77 (21%)		192 (52.5%)	174 (47.5%)	
Living environment			0.898			0.279
Urban	137 (77.8%)	39 (22.2%)		81 (46.0%)	95 (54.0%)	
Suburban	114 (77.6%)	33 (22.4%)		77 (52.7%)	69 (47.3%)	
Rural	188 (79.3%)	49 (20.7%)		127 (53.6%)	110 (46.4%)	

**Table 2 vaccines-12-00602-t002:** Decision-making about vaccination against COVID-19 with primary and booster doses among nurses in primary healthcare in Slovenia according to comorbidities, unhealthy lifestyle and habits, and self-rated health.

	Vaccinated with Primary Doses (*n* = 560)		Pearson Chi-Square Sig. (2-Sided)	Vaccinated with Booster Doses (*n* = 560)		Pearson Chi-Square Sig. (2-Sided)
	Yes	No		Yes	No	
	*n* (%)	*n* (%)		*n* (%)	*n* (%)	
Comorbidities						
Chronic respiratory disease	17 (81%)	4 (19%)	0.771	9 (42.9%)	12 (57.1%)	0.448
Chronic heart disease	15 (93.8%)	1 (6.3%)	0.130	12 (75%)	4 (25%)	0.051
Diabetes mellitus	2 (66.7%)	1 (33.3%)	0.621	1 (33.3%)	2 (66.7%)	0.540
Chronic kidney or liver disease	6 (85.7%)	1 (14.3%)	0.636	(57.1%)	3 (42.9%)	0.743
Another chronic health issue	41 (75.9%)	13 (24.1%)	0.643	30 (55.6%)	24 (44.4%)	0.480
Immunocompromised	10 (71.4%)	4 (28.6%)	0.521	6 (42.9%)	8 (57.1%)	0.538
Unhealthy lifestyle and habits						
A smoker	81 (82.7%)	17 (7.3%)	0.266	48 (49.0%)	50 (51.0%)	0.717
Being overweight (BMI ≥ 30)	75 (88.2%)	10 (11.8%)	0.017	52 (61.2%)	33 (38.8%)	0.034
Physically less active	112 (86.2%)	18 (13.8%)	0.015	72 (55.4%)	58 (44.6%)	0.215
Lacks fruit and veg in the diet	64 (80%)	16 (20%)	0.718	34 (42.5%)	46 (57.5%)	0.115
Self-rated health						
Considered to be healthy	343 (78.3%)	94 (21.5%)	0.916	219 (50.2%)	217 (49.8%)	0.502
Number of comorbidities						
1 comorbidity *	83 (78.3%)	23 (21.7%)	0.866	59 (55.7%)	47 (44.3%)	0.266
≥2 comorbidities	13 (76.5%)	4 (23.5%)	0.866	7 (41.2%)	10 (58.8%)	0.266

* Comorbidities listed in the survey: asthma or other chronic respiratory disease, chronic heart disease, diabetes, chronic kidney or liver disease, immunocompromised condition, or another chronic health issue.

**Table 3 vaccines-12-00602-t003:** Cramer’s V coefficient for vaccination with primary doses.

Variable	Cramer’s V Coefficient	Explanation
Gender	0.022	The value of Cramer’s V is 0.022, indicating a very weak and statistically insignificant (Sig. = 0.597) association between the variables “vaccinated against COVID-19 with two doses as per the primary scheme” and “Gender”.
Age	0.193	The value of Cramer’s V is 0.193, indicating a weak but statistically significant (Sig. < 0.001) association between the variables “vaccinated against COVID-19 following the primary scheme with two doses” and “Age”.
Body weight ≥ 30 BMI	0.102	The value of Cramer’s V is 0.102, which suggests a weak yet statistically significant (Sig. = 0.017) association between the variables “vaccinated against COVID-19 with two doses as per the primary scheme” and “excessive body weight, more than 30 BMI”.
Insufficient physical activity	0.104	The value of Cramer’s V is 0.104, which suggests a weak but statistically significant (Sig. = 0.015) association between the variables “vaccinated against COVID-19 following the primary scheme with two doses” and “insufficient physical activity, less than 150 min per week”.

**Table 4 vaccines-12-00602-t004:** Cramer’s V coefficient for vaccination with booster doses.

Variable	Cramer’s V Coefficient	Explanation
Gender	0.038	The value of Cramer’s V is 0.038, which suggests a very weak and statistically non-significant (Sig. = 0.375) association between the variables “vaccinated against COVID-19 with a booster dose” and “Gender”.
Age	0.258	The value of Cramer’s V is 0.258, which indicates a moderate and statistically significant (Sig. < 0.001) association between the variables “vaccinated against COVID-19 with a booster dose” and “Age”.
Body weight ≥ 30 BMI	0.090	The value of Cramer’s V is 0.090, which suggests a weak but statistically significant (Sig. = 0.034) association between the variables “vaccinated against COVID-19 with a booster dose” and “excessive body weight, more than 30 BMI”.
Insufficient physical activity	0.053	The value of Cramer’s V is 0.053, which indicates a very weak and statistically non-significant (Sig. = 0.215) association between the variables “vaccinated against COVID-19 with a booster dose” and “insufficient physical activity, less than 150 min per week”.

**Table 5 vaccines-12-00602-t005:** Logistic regression model for basic vaccination scheme with COVID-19 vaccine in primary care nurses.

Variable	B	S.E.	Wald	df	Sig.	Exp(B)
Gender	0.214	0.397	0.291	1	0.589	1.239
Age	0.894	0.230	15.073	1	0.000	2.446
Education	0.234	0.241	0.950	1	0.330	1.264
Living environment	0.058	0.132	0.191	1	0.662	1.059
Chronic respiratory disease	0.274	0.775	0.125	1	0.724	1.315
Chronic heart disease	1.576	1.170	1.815	1	0.178	4.835
Diabetes mellitus	−1.199	1.446	0.687	1	0.407	0.302
Chronic kidney or liver disease	0.489	1.233	0.157	1	0.692	1.630
Another chronic health issue	−0.085	0.620	0.019	1	0.892	0.919
Immunocompromised	−0.236	1.001	0.056	1	0.814	0.790
A smoker	0.334	0.318	1.100	1	0.294	1.397
Being overweight (BMI ≥ 30)	0.512	0.382	1.795	1	0.180	1.669
Physically less active	0.648	0.323	4.026	1	0.045	1.911
Lacks fruit and veg in the diet	−0.063	0.354	0.031	1	0.859	0.939
Considered to be healthy	−0.925	1.161	0.635	1	0.425	0.396
1 comorbidity	−0.241	0.491	0.242	1	0.623	0.786
≥2 comorbidities	−0.376	1.384	0.074	1	0.786	0.687

**Table 6 vaccines-12-00602-t006:** Logistic regression model for booster dose with COVID-19 vaccine in primary care nurses.

Variable	B	S.E.	Wald	df	Sig.	Exp(B)
Gender	0.742	0.337	4.838	1	0.028	2.101
Age	1.033	0.190	29.419	1	0.000	2.810
Education	0.272	0.203	1.808	1	0.179	1.313
Living environment	0.190	0.110	2.973	1	0.085	1.209
Chronic respiratory disease	0.064	0.652	0.010	1	0.922	1.066
Chronic heart disease	1.628	0.749	4.722	1	0.030	5.096
Diabetes mellitus	−0.936	1.446	0.419	1	0.517	0.392
Chronic kidney or liver disease	0.617	0.942	0.429	1	0.513	1.853
Another chronic health issue	0.752	0.537	1.961	1	0.161	2.120
Immunocompromised	0.237	0.879	0.073	1	0.787	1.268
A smoker	−0.004	0.252	0.000	1	0.986	0.996
Being overweight (BMI ≥ 30)	0.293	0.276	1.127	1	0.288	1.341
Physically less active	0.354	0.247	2.050	1	0.152	1.425
Lacks fruit and veg in the diet	−0.539	0.294	3.370	1	0.046	0.583
Considered to be healthy	−0.468	0.739	0.402	1	0.526	0.626
1 comorbidity	−0.394	0.421	0.876	1	0.349	0.674
≥2 comorbidities	−1.227	1.187	1.067	1	0.302	0.293

## Data Availability

Data are available at request from data curator Z.P.

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
