# Peer review of "Factors Influencing COVID-19 Vaccination among Primary Healthcare Nurses in the Pandemic and Post-Pandemic Period: Cross-Sectional Study"

_vaccines, 2024, doi:10.3390/vaccines12060602_

Round 1

Reviewer 1 Report

Comments and Suggestions for Authors

The article is very long, and should be curtailed.

The figures do not add important information.

Some sentences are not clear, see line 152.

The format of the tables is not easy to understand.

Comments on the Quality of English Language

The article is very long, and should be curtailed.

The figures do not add important information.

Some sentences are not clear, see line 152.

The format of the tables is not easy to understand.

Author Response

Comments and Suggestions for Authors

We would like to thank the reviewer for careful review of our article and useful comments that will help to improve it.

The article is very long, and should be curtailed. The figures do not add important information.

Following your suggestion, we removed Figures 1 and 2 which do not contribute additional information and removed from the text the part that is connected to Figures.

Some sentences are not clear, see line 152.

We have changed the sentence in line 152 to make it clearer. The previous sentence was: “The call for participation garnered 913 clicks. However, not all participants completed the survey, and some were inadequately filled out, resulting in 560 questionnaires being included in the final analysis.”

was changed to: We invited 3130 nurses to participate in web-based anonymous survey. We noticed that 913 nurses opened the survey but not all of them took the decision to participate. Some nurses fulfilled the questionnaire inadequately and were omitted from the analysis. There were 560 questionnaires adequately fulfilled and available for the final analysis.    

The format of the tables is not easy to understand.

We have included in new version additional tables (Table 1 was corrected, Table 5 and 6 added) to be   clear and significantly more comprehensible.

Reviewer 2 Report

Comments and Suggestions for Authors

- The manuscript lacks a thorough discussion on the potential for self-report and recall bias, which are significant considerations given the study design involving survey data. The authors should discuss the likelihood and impact of these biases and mention any steps taken to mitigate them. It is important to address the bias in the study as it assesses how trustworthy the result is. While the study presents valuable insights into vaccine hesitancy among primary healthcare nurses, it's essential to acknowledge and address potential biases inherent in the study design. Specifically, self-report and recall biases are significant concerns in survey-based research, particularly when assessing sensitive topics such as vaccination beliefs and behaviors.

- After mentioning the limitation at the end of the discussion section, it would be good to explore the future direction. By failing to explore the next steps or areas for further investigation, it misses an opportunity to guide future inquiry and action in addressing COVID-19 vaccine hesitancy.

- In Figures 1 and 2, the Y-axis legend should be labeled properly.

Author Response

We would like to thank the reviewer for careful review of our article and useful comments that will help to improve it.

  1. The manuscript lacks a thorough discussion on the potential for self-report and recall bias, which are significant considerations given the study design involving survey data. The authors should discuss the likelihood and impact of these biases and mention any steps taken to mitigate them. It is important to address the bias in the study as it assesses how trustworthy the result is. While the study presents valuable insights into vaccine hesitancy among primary healthcare nurses, it's essential to acknowledge and address potential biases inherent in the study design. Specifically, self-report and recall biases are significant concerns in survey-based research, particularly when assessing sensitive topics such as vaccination beliefs and behaviors.

We added the following text to the limitation paragraph of Discussion section:

Self-reporting is a common approach for collecting data in public health research. Self-reported data might be unreliable and threatened by self-reporting bias. Bias can arise from social desirability and recall period, among some other biases (Alaa Althubaiti. Information bias in health research: definition, pitfalls, and adjustment methods. J Multidiscip Healthc. 2016; 9: 211–217). The participants in present study could erroneously delivered responses that depend on his/her ability to recall past event e.g. vaccination with COVID-19 vaccine. As the study population consisted of nurses we believe that the probability to mistakenly remember and to report the vaccination status was highly unlikely but still possible and might resulted in incorrect classification of few cases to one of the groups analyzed. For most of the questions, the length of recall period was short, which again lowers the probability of the recall bias. Nevertheless, we are aware that overcoming recall bias can be difficult in practice. Self-reporting data can also be affected by an external bias caused by social desirability. We did not measure internal or external validity of the data which is one of the limitations of our study. The questionnaire was anonymous and the identification of participants not possible, we expected that their answers were trustworthy.  

- After mentioning the limitation at the end of the discussion section, it would be good to explore the future direction. By failing to explore the next steps or areas for further investigation, it misses an opportunity to guide future inquiry and action in addressing COVID-19 vaccine hesitancy.

Exploring future directions for addressing COVID-19 vaccine hesitancy among nurses is crucial for guiding further research and action in this area. Here are some potential avenues for investigation and action. We added the following text as the last paragraph of Discussion section:

Further in-depth quality studies are needed to look deeper into the reasons behind vaccine hesitancy among nurses through face-to-face interviews or focus groups to understand their concerns, beliefs, and attitudes towards COVID-19 vaccines and tackle cultural and contextual factors. There is a need to develop tailored educational programs for this professional group to address common misconceptions, provide accurate information about vaccine safety and efficacy, and highlight the importance of vaccination in protecting both individual and public health. Educational programs should be regularly evaluated for the effectiveness, analyzing vaccination rates before and after implementation. More research is warranted on the potential role of policy interventions in promoting workplace vaccination policies, offering incentives for vaccination, or addressing systemic barriers to vaccine access. An interesting concept in the nowadays society is to encourage influential nurses to serve as vaccine advocates and role models within their communities to combat misinformation and build trust in vaccines through social networks. Lastly, developing effective strategies for addressing COVID-19 vaccine hesitancy among nurses and promoting vaccination as a critical public health measure should be recognized and supported by policymakers.

We would like to thank the reviewer for careful review of our article and useful comments that will help to improve it.

  1. The manuscript lacks a thorough discussion on the potential for self-report and recall bias, which are significant considerations given the study design involving survey data. The authors should discuss the likelihood and impact of these biases and mention any steps taken to mitigate them. It is important to address the bias in the study as it assesses how trustworthy the result is. While the study presents valuable insights into vaccine hesitancy among primary healthcare nurses, it's essential to acknowledge and address potential biases inherent in the study design. Specifically, self-report and recall biases are significant concerns in survey-based research, particularly when assessing sensitive topics such as vaccination beliefs and behaviors.

We added the following text to the limitation paragraph of Discussion section:

Self-reporting is a common approach for collecting data in public health research. Self-reported data might be unreliable and threatened by self-reporting bias. Bias can arise from social desirability and recall period, among some other biases (Alaa Althubaiti. Information bias in health research: definition, pitfalls, and adjustment methods. J Multidiscip Healthc. 2016; 9: 211–217). The participants in present study could erroneously delivered responses that depend on his/her ability to recall past event e.g. vaccination with COVID-19 vaccine. As the study population consisted of nurses we believe that the probability to mistakenly remember and to report the vaccination status was highly unlikely but still possible and might resulted in incorrect classification of few cases to one of the groups analyzed. For most of the questions, the length of recall period was short, which again lowers the probability of the recall bias. Nevertheless, we are aware that overcoming recall bias can be difficult in practice. Self-reporting data can also be affected by an external bias caused by social desirability. We did not measure internal or external validity of the data which is one of the limitations of our study. The questionnaire was anonymous and the identification of participants not possible, we expected that their answers were trustworthy.  

- After mentioning the limitation at the end of the discussion section, it would be good to explore the future direction. By failing to explore the next steps or areas for further investigation, it misses an opportunity to guide future inquiry and action in addressing COVID-19 vaccine hesitancy.

Exploring future directions for addressing COVID-19 vaccine hesitancy among nurses is crucial for guiding further research and action in this area. Here are some potential avenues for investigation and action. We added the following text as the last paragraph of Discussion section:

Further in-depth quality studies are needed to look deeper into the reasons behind vaccine hesitancy among nurses through face-to-face interviews or focus groups to understand their concerns, beliefs, and attitudes towards COVID-19 vaccines and tackle cultural and contextual factors. There is a need to develop tailored educational programs for this professional group to address common misconceptions, provide accurate information about vaccine safety and efficacy, and highlight the importance of vaccination in protecting both individual and public health. Educational programs should be regularly evaluated for the effectiveness, analyzing vaccination rates before and after implementation. More research is warranted on the potential role of policy interventions in promoting workplace vaccination policies, offering incentives for vaccination, or addressing systemic barriers to vaccine access. An interesting concept in the nowadays society is to encourage influential nurses to serve as vaccine advocates and role models within their communities to combat misinformation and build trust in vaccines through social networks. Lastly, developing effective strategies for addressing COVID-19 vaccine hesitancy among nurses and promoting vaccination as a critical public health measure should be recognized and supported by policymakers.

We would like to thank the reviewer for careful review of our article and useful comments that will help to improve it.

  1. The manuscript lacks a thorough discussion on the potential for self-report and recall bias, which are significant considerations given the study design involving survey data. The authors should discuss the likelihood and impact of these biases and mention any steps taken to mitigate them. It is important to address the bias in the study as it assesses how trustworthy the result is. While the study presents valuable insights into vaccine hesitancy among primary healthcare nurses, it's essential to acknowledge and address potential biases inherent in the study design. Specifically, self-report and recall biases are significant concerns in survey-based research, particularly when assessing sensitive topics such as vaccination beliefs and behaviors.

We added the following text to the limitation paragraph of Discussion section:

Self-reporting is a common approach for collecting data in public health research. Self-reported data might be unreliable and threatened by self-reporting bias. Bias can arise from social desirability and recall period, among some other biases (Alaa Althubaiti. Information bias in health research: definition, pitfalls, and adjustment methods. J Multidiscip Healthc. 2016; 9: 211–217). The participants in present study could erroneously delivered responses that depend on his/her ability to recall past event e.g. vaccination with COVID-19 vaccine. As the study population consisted of nurses we believe that the probability to mistakenly remember and to report the vaccination status was highly unlikely but still possible and might resulted in incorrect classification of few cases to one of the groups analyzed. For most of the questions, the length of recall period was short, which again lowers the probability of the recall bias. Nevertheless, we are aware that overcoming recall bias can be difficult in practice. Self-reporting data can also be affected by an external bias caused by social desirability. We did not measure internal or external validity of the data which is one of the limitations of our study. The questionnaire was anonymous and the identification of participants not possible, we expected that their answers were trustworthy.  

- After mentioning the limitation at the end of the discussion section, it would be good to explore the future direction. By failing to explore the next steps or areas for further investigation, it misses an opportunity to guide future inquiry and action in addressing COVID-19 vaccine hesitancy.

Exploring future directions for addressing COVID-19 vaccine hesitancy among nurses is crucial for guiding further research and action in this area. Here are some potential avenues for investigation and action. We added the following text as the last paragraph of Discussion section:

Further in-depth quality studies are needed to look deeper into the reasons behind vaccine hesitancy among nurses through face-to-face interviews or focus groups to understand their concerns, beliefs, and attitudes towards COVID-19 vaccines and tackle cultural and contextual factors. There is a need to develop tailored educational programs for this professional group to address common misconceptions, provide accurate information about vaccine safety and efficacy, and highlight the importance of vaccination in protecting both individual and public health. Educational programs should be regularly evaluated for the effectiveness, analyzing vaccination rates before and after implementation. More research is warranted on the potential role of policy interventions in promoting workplace vaccination policies, offering incentives for vaccination, or addressing systemic barriers to vaccine access. An interesting concept in the nowadays society is to encourage influential nurses to serve as vaccine advocates and role models within their communities to combat misinformation and build trust in vaccines through social networks. Lastly, developing effective strategies for addressing COVID-19 vaccine hesitancy among nurses and promoting vaccination as a critical public health measure should be recognized and supported by policymakers.

We would like to thank the reviewer for careful review of our article and useful comments that will help to improve it.

  1. The manuscript lacks a thorough discussion on the potential for self-report and recall bias, which are significant considerations given the study design involving survey data. The authors should discuss the likelihood and impact of these biases and mention any steps taken to mitigate them. It is important to address the bias in the study as it assesses how trustworthy the result is. While the study presents valuable insights into vaccine hesitancy among primary healthcare nurses, it's essential to acknowledge and address potential biases inherent in the study design. Specifically, self-report and recall biases are significant concerns in survey-based research, particularly when assessing sensitive topics such as vaccination beliefs and behaviors.

We added the following text to the limitation paragraph of Discussion section:

Self-reporting is a common approach for collecting data in public health research. Self-reported data might be unreliable and threatened by self-reporting bias. Bias can arise from social desirability and recall period, among some other biases (Alaa Althubaiti. Information bias in health research: definition, pitfalls, and adjustment methods. J Multidiscip Healthc. 2016; 9: 211–217). The participants in present study could erroneously delivered responses that depend on his/her ability to recall past event e.g. vaccination with COVID-19 vaccine. As the study population consisted of nurses we believe that the probability to mistakenly remember and to report the vaccination status was highly unlikely but still possible and might resulted in incorrect classification of few cases to one of the groups analyzed. For most of the questions, the length of recall period was short, which again lowers the probability of the recall bias. Nevertheless, we are aware that overcoming recall bias can be difficult in practice. Self-reporting data can also be affected by an external bias caused by social desirability. We did not measure internal or external validity of the data which is one of the limitations of our study. The questionnaire was anonymous and the identification of participants not possible, we expected that their answers were trustworthy.  

- After mentioning the limitation at the end of the discussion section, it would be good to explore the future direction. By failing to explore the next steps or areas for further investigation, it misses an opportunity to guide future inquiry and action in addressing COVID-19 vaccine hesitancy.

Exploring future directions for addressing COVID-19 vaccine hesitancy among nurses is crucial for guiding further research and action in this area. Here are some potential avenues for investigation and action. We added the following text as the last paragraph of Discussion section:

Further in-depth quality studies are needed to look deeper into the reasons behind vaccine hesitancy among nurses through face-to-face interviews or focus groups to understand their concerns, beliefs, and attitudes towards COVID-19 vaccines and tackle cultural and contextual factors. There is a need to develop tailored educational programs for this professional group to address common misconceptions, provide accurate information about vaccine safety and efficacy, and highlight the importance of vaccination in protecting both individual and public health. Educational programs should be regularly evaluated for the effectiveness, analyzing vaccination rates before and after implementation. More research is warranted on the potential role of policy interventions in promoting workplace vaccination policies, offering incentives for vaccination, or addressing systemic barriers to vaccine access. An interesting concept in the nowadays society is to encourage influential nurses to serve as vaccine advocates and role models within their communities to combat misinformation and build trust in vaccines through social networks. Lastly, developing effective strategies for addressing COVID-19 vaccine hesitancy among nurses and promoting vaccination as a critical public health measure should be recognized and supported by policymakers.

Reviewer 3 Report

Comments and Suggestions for Authors

This is well designed and conducted study aiming to identify decision factors of anti-covid vaccination aiming primary healthcare nurses. Survey-based study was conducted in Slovenia in 2023 and included randomized selection of 560 nurses representing at least half of the health centers from each statistical region in the country. There were no statistically significant differences in the decision to vaccinate against COVID-19 with either the primary or booster dose when considering gender and level of education which align with previous studies. They found, however, that more men than women, older individuals aged 41 years or those with more than secondary education, and those living in suburban areas chose the booster dose compared to the primary vaccination. Among those who indicated having any comorbidity, those with chronic heart disease were most likely to opt for the booster vaccination, surprisingly, the least likely were those with asthma or other chronic respiratory diseases. Strength of the study are statistical approach and randomized selection of participants; results are objectively interpreted and discussed. Reported observation confirm similar reports on decision influencing Covid-19 vaccination and extend findings to relatively homogenous group of healthcare workers. Among probable queries please discuss whether conclusion that men were more likely to receive booster is valid since this group constituted only 5% of surveyed nurses.

Author Response

We would like to thank the reviewer for careful review of our article and useful comments that will help to improve it.

This is well designed and conducted study aiming to identify decision factors of anti-covid vaccination aiming primary healthcare nurses. Survey-based study was conducted in Slovenia in 2023 and included randomized selection of 560 nurses representing at least half of the health centers from each statistical region in the country. There were no statistically significant differences in the decision to vaccinate against COVID-19 with either the primary or booster dose when considering gender and level of education which align with previous studies. They found, however, that more men than women, older individuals aged 41 years or those with more than secondary education, and those living in suburban areas chose the booster dose compared to the primary vaccination. Among those who indicated having any comorbidity, those with chronic heart disease were most likely to opt for the booster vaccination, surprisingly, the least likely were those with asthma or other chronic respiratory diseases. Strength of the study are statistical approach and randomized selection of participants; results are objectively interpreted and discussed. Reported observation confirm similar reports on decision influencing Covid-19 vaccination and extend findings to relatively homogenous group of healthcare workers. Among probable queries please discuss whether conclusion that men were more likely to receive booster is valid since this group constituted only 5% of surveyed nurses.

Thank you for your suggestion. The following text was added to the Discussion section (limitations of the study):

The gender imbalance in our study is similar to that in comparable studies, as there are significantly more women than men employed in healthcare, especially in the profession of nurse. The small number of men calls into question whether we can conclude that more male than female nurses opted for booster doses. This inherent obstacle i.e. the gender imbalance could not be solved, but the result is in line with similar studies, which found that men were more willing to be vaccinated, regardless of education level.

Reviewer 4 Report

Comments and Suggestions for Authors

Dear Editor, Dear Authors,

Thank you for the opportunity to review the manuscript entitled Decision Factors for COVID-19 Vaccination among Primary Healthcare Nurses in the Pandemic and Post-Pandemic Period: Cross-sectional Study. The paper addressed an interesting topic, which was rarely investigated in Central European countries. The study results should shed light on determinants of willingness and hesitancy to take newly develop vaccines by nurses, considering the experience of COVID-19. Manuscript, although interesting, suffers several issues which cause it is not acceptable for publication. The majority are as follows:

*the content of the manuscript missed several elements which should be presented. I strongly recommend the use of STROBE Checklist to improve the scientific soundness  of the paper (https://www.strobe-statement.org/)

Abstract

*present some numbered results which support the conclusions.

*present the conclusions showing the nature of the association (direction) but not only ‘differences were observed’

Introduction:

*here are several theories to explain determinants and impact of different factors on decision making including the participation in health programs. Present and refer to some models  which may/might explain the role of factors considered in the presented study

Purpose:

*The authors mention “the study sought to respond to questions about the explanatory power of …”. There is no, however, analyses which address that point, no results, no conclusions

Materials & Methods

*authors decide to select health centers (page.3-line.101) which violates random sampling, explain the rationale and discuss under study limitations

*explain what type of probability sampling have you used. It is not clear whether it was a weighting sampling or not, how the weights were calculated

*provide information on the sample size determination and what assumptions were used to calculate

*provide, please, the response rate, steps undertaken to verify the representativeness of the study sample

*statistical analysis has been performed inadequately, as the chi-squared test does not enable control for covariates. Making conclusions on the basis of that test with the presence of differences in the distribution of other possible determinants is a methodologically incorrect

*although the Cramer’s  V is a measure of the strength of the association between two nominal variables its use is very limited (there is no control for other co-variables, it is a heavily biased estimator and tends to overestimate the strength of association) and currently it is hardly to accept conclusion making based on that statistic.

*Descriptions provided in tables 3 & 4, although correct, don’t show the nature of the association … e.g. how gender is associated with primary doses?

*I suggest running the structural equation modeling to analyze the complexity of the process or at least uni- and next multivariable logistic regression to control for possible confounders/covariates

*the rationale for categorization of continuous variables is not presented

Results

*calculation and presentation % in tables is mistaken. E.g. if you compare vaccinated Yes/No and the proportion of men/women across vaccinated you should present the proportion of men out of the whole group vaccinated vs. the proportion of men out of the whole group of non-vaccinated.

% should be corrected for all presented results.

*the limitations of the study are poorly presented and discussed

Conclusions

*”Conclusion is a judgment or decision reached by reasoning, it should sum up the arguments and present the final deduction based on the results observed.” The proposed form of conclusion does not fulfill that criteria and requires to be corrected.

Comments on the Quality of English Language

In my opinion the manuscript requires English language revision and corrections by native speaker. There are several grammar issues and inproper preposition use, which, if corrected, improve the readability of the manuscript.

Author Response

We would like to thank the reviewer for careful review of our article and useful comments that will help to improve it.

Dear Editor, Dear Authors,

Thank you for the opportunity to review the manuscript entitled Decision Factors for COVID-19 Vaccination among Primary Healthcare Nurses in the Pandemic and Post-Pandemic Period: Cross-sectional Study. The paper addressed an interesting topic, which was rarely investigated in Central European countries. The study results should shed light on determinants of willingness and hesitancy to take newly develop vaccines by nurses, considering the experience of COVID-19. Manuscript, although interesting, suffers several issues which cause it is not acceptable for publication. The majority are as follows:

*the content of the manuscript missed several elements which should be presented. I strongly recommend the use of STROBE Checklist to improve the scientific soundness  of the paper (https://www.strobe-statement.org/)

We would like to thank the reviewer for the useful suggestion. We used STROBE Checklist to improve the paper. 

Abstract

*present some numbered results which support the conclusions.

*present the conclusions showing the nature of the association (direction) but not only ‘differences were observed’

The number have been included in the abstract and the nature of the direction association described.

The sentence: “Statistically, individuals with two unhealthy lifestyle factors, specifically excessive body weight and insufficient physical activity, were more likely to opt for vaccination following the primary scheme”. was changed to: “Primary care nurses who were ≥41 years of age, physically less active and those who were over-weight opted statistically significant more often for primary scheme vaccination (p= 0.00, 0.015 and 0.017, respectively).”

The sentence: “Significant differences were observed for booster doses with age and excessive body weight” was changed to: “Vaccinated with booster doses were significantly more often overweight (p=0.034) and were ≥41 year of age (p=0.000).”   

Introduction:

*here are several theories to explain determinants and impact of different factors on decision making including the participation in health programs. Present and refer to some models  which may/might explain the role of factors considered in the presented study

World Health Organization (WHO) defines socio-demographic characteristics as factors influencing health, whereby the decision to vaccinate can also be linked to some socio-demographic factors such as age, gender, education, living environment etc. [11].

According to the Health Belief Model, having health risk is one of the decision-making factor. Janz and Becker calculated significance ratios indicating how often each factor was statistically significant in 46 studies. The impact of susceptibility for infection, seriousness of the disease, perceived benefits and severity on vaccine acceptance were 81%, 65%, 78% and 89%, respectively [12].

World Health Organization. Social determinants of health. https://www.who.int/health-topics/social-determinants-of-health#tab=tab_1

Janz, N. K. in Becker, M. H., 1984. The health belief model: a decade later. Health Education Quarterly, spring 1984;11(1), pp. 1–47.

Purpose:

*The authors mention “the study sought to respond to questions about the explanatory power of …”. There is no, however, analyses which address that point, no results, no conclusions

The sentence has been left out.

Materials & Methods

*authors decide to select health centers (page.3-line.101) which violates random sampling, explain the rationale and discuss under study limitations

The health centers were selected by drawing lots (see below).

*explain what type of probability sampling have you used. It is not clear whether it was a weighting sampling or not, how the weights were calculated

There was no weighting and weights were not calculated (see below).

*provide information on the sample size determination and what assumptions were used to calculate

About 6,000 nurses are employed in primary care health centers in Slovenia. From each statistical region in the country, we selected by drawing lots half of the health centers in every statistical region without weighting. We selected 32 medical centers (out of a total of 66) with a total number of 3130 nurses working in primary care and invited them to take a part in cross-sectional study.

*provide, please, the response rate, steps undertaken to verify the representativeness of the study sample

The primary care nurses were invited twice to participate in the study but still the response rate was less than 20 %. The low response rate and questionable representativeness was discussed as one of the limitations of the study.

*statistical analysis has been performed inadequately, as the chi-squared test does not enable control for covariates. Making conclusions on the basis of that test with the presence of differences in the distribution of other possible determinants is a methodologically incorrect

Logistic regression model was constructed and added to Methods and Results section.

Added to Methods:

We used logistic regression to analyze the data. The dependent variables were binary (whether the individual was vaccinated with the basic scheme or not and whether the individual was vaccinated with the booster dose or not). We estimated the probability to get vaccinated based on independent variables (socio-demographic data, health conditions and life style).

Maximum likelihood was used to estimate the coefficients of the regression models. The statistical significance of the coefficients was checked using the Wald test, using a value of p < 0.05 as the significance threshold. The coefficients were interpreted using an exponent (Exp(B)) representing the likelihood ratio.

Data analysis was performed using the statistical software package SPSS (version 29.0).

Added to Results

Multiple regression model for basic vaccination scheme

For the basic vaccination scheme, the age was the most important variable, as with each additional year of age, the probability of vaccination increased by a factor 2.446 (B=0.894; Sig.<0.001) (Table 5). Physical activity also had a positive influence, with individuals who are less physically active had a 1.911 times higher probability of vaccination (B=0.648; Sig=0.045). Other variables, such as gender, education, environment of permanent residence and the presence of other chronic diseases, did not have a statistically significant effect on the decision to be vaccinated.

Multiple regression model for booster dose

For the booster dose, age was also the most important factor, as the probability of vaccination increased by a factor 2.810 with each additional year of age (B=1.033; Sig.<0.001) (Table 6). Gender was also a significant factor, as men were 2.101 times more likely to get booster dose (B=0.742; Sig=0.028). The presence of chronic heart disease increased the probability of vaccination by a factor  5.096 (B=1.628; Sig=0.030). Dietary habits, especially consumption of less than 400 grams of fruits and vegetables per day, had a negative effect on the decision to vaccinate, decreasing the probability of getting booster dose by a factor 0.583 (B=-0.539; Sig=0.046). Other variables in the model did not have a statistically significant effect on the decision to get booster dose.

*although the Cramer’s  V is a measure of the strength of the association between two nominal variables its use is very limited (there is no control for other co-variables, it is a heavily biased estimator and tends to overestimate the strength of association) and currently it is hardly to accept conclusion making based on that statistic.

We are well aware of the limitations of the Cramer’s V but the conclusions were not based on Cramer’s V as a measure of the strength of the association.  

*Descriptions provided in tables 3 & 4, although correct, don’t show the nature of the association … e.g. how gender is associated with primary doses?

As shown in Table 3, gender is weakly and insignificantly associated with primary vaccination scheme acceptance.

*I suggest running the structural equation modeling to analyze the complexity of the process or at least uni- and next multivariable logistic regression to control for possible confounders/covariates

The multiple regression model was added to Results section (and to the Methods).

*the rationale for categorization of continuous variables is not presented

There is one continuous variable which was categorized in two categories but left continuous in the multiple regression model.

Results

*calculation and presentation % in tables is mistaken. E.g. if you compare vaccinated Yes/No and the proportion of men/women across vaccinated you should present the proportion of men out of the whole group vaccinated vs. the proportion of men out of the whole group of non-vaccinated.

% should be corrected for all presented results.

The tables have been corrected as suggested.

*the limitations of the study are poorly presented and discussed

More limitations of the study were added:

- low response rate as pointed by the reviewer;

- gender imbalance;

- self-administrated questionnaire which might create certain biases.

Conclusions

*”Conclusion is a judgment or decision reached by reasoning, it should sum up the arguments and present the final deduction based on the results observed.” The proposed form of conclusion does not fulfill that criteria and requires to be corrected.

We corrected the Conclusions according to reviewer’s suggestions:

This study revealed that different factors are important in the decision to vaccinate against COVID-19 with the first dose and booster dose. Based on the findings, we can conclude that older individuals are more in favor to be vaccinated against COVID-19, which applies to primary scheme and booster dose. It was also found that being less physical activity increased the probability of vaccination with the basic scheme, while eating habits had significant influence on booster dose.

Round 2

Reviewer 1 Report

Comments and Suggestions for Authors

The revised article can be published.

Comments on the Quality of English Language

OK

Reviewer 4 Report

Comments and Suggestions for Authors

Dear Editor, Dear Authors,

Although the novelty of the topic presented in the submitted manuscript is somehow limited, the paper provides some evidence on determinants of willingness and hesitancy to take newly develop vaccines by nurses in the Central European region. I believe, this knowledge should be used by policy makers, therefore the manuscript holds promise to be cited.

The revised version clarifies the major issues noted in the first revision, and in the current form seems to fulfill the criteria to be published.